# Advanced, Innovative AIoT and Edge Computing for Unmanned Vehicle Systems in Factories

**Yen-Hui Kuo \* and Eric Hsiao-Kuang Wu**

Department of Computer Science and Information Engineering, National Central University, Jhongli 32001, Taiwan; hsiao@csie.ncu.edu.tw
\* Correspondence: kyh97730@gmail.com

**Abstract:** Post-COVID-19, there are frequent manpower shortages across industries. Many factories pursuing future technologies are actively developing smart factories and introducing automation equipment to improve factory manufacturing efficiency. However, the delay and unreliability of existing wireless communication make it difficult to meet the needs of AGV navigation. Selecting the right sensor, reliable communication, and navigation control technology remains a challenging issue for system integrators. Most of today's unmanned vehicles use expensive sensors or require new infrastructure to be deployed, impeding their widespread adoption. In this paper, we have developed a self-learning and efficient image recognition algorithm. We developed an unmanned vehicle system that can navigate without adding any specialized infrastructure, and tested it in the factory to verify its usability. The novelties of this system are that we have developed an unmanned vehicle system without any additional infrastructure, and we developed a rapid image recognition algorithm for unmanned vehicle systems to improve navigation safety. The core contribution of this system is that the system can navigate smoothly without expensive sensors and without any additional infrastructure. It can simultaneously support a large number of unmanned vehicle systems in a factory.

**Keywords:** artificial intelligence; Internet of Things; edge computing; smart factories; image recognition; unmanned vehicle





## 1. Introduction

In order to enhance competitiveness, companies seek various methods of enhancing manufacturing efficiency, increasing flexibility, and reducing costs. In the context of Industry 4.0 [1], intelligent automatic guided vehicles in factories minimize the manpower required to conduct low-level work [2]. AGVs are commonly used in factories to move materials. Due to their widespread use in smart manufacturing, the global AGV hardware, software, and services market was worth around USD 4.8 billion in 2022, with a projected revenue CAGR of around 10% [3]. In the post-pandemic era, there are frequent shortages of manpower in various industries. Smart factories are thus being developed, and automation equipment has been introduced to improve efficiency. However, most traditional AGV navigation hardware comprises an on-board processor that has limited processing capabilities, meaning that it may not be able to perform complex calculations, hindering the development of related systems. With the continuous development of artificial intelligence, wireless networks, IoT, and edge computing technologies, the connection of networked devices enables "things" to communicate, exchange data, and improve operational efficiency [4,5]. Edge computing [6,7] is a next-generation wireless network and artificial intelligence IoT solution that can be used as an extension of cloud computing to support the supply of continuous services from the cloud and reduce a network's bandwidth requirements. The development of advanced technologies will lead to the development of new business practices in smart factories. For smart factories, there is an additional cost to installing additional hardware of any kind. There are aisles in the factory. Colored lines separate

the work area, product area, and aisles. These colored lines can be used for unmanned vehicle system navigation. This is a great solution for the factory of the future (FoF) as the existing infrastructure of the factory can be used without installing any kind of additional hardware infrastructure. In this paper, we try to integrate innovative vision technology, AI, IoT, and edge computing technology to develop unmanned vehicle systems that do not require additional hardware infrastructure. We used a real unmanned vehicle system to validate this solution.

The remainder of this paper is organized as follows. In Section 2, the materials and methods are introduced. Section 3 introduces our experimental results. In Section 4, our methods and results are discussed. We conclude this paper in Section 5.

### 1.1. Related Work and Background Knowledge

Unmanned vehicle systems are used in factories to deliver materials. Factories contain people, equipment, materials, products, etc. Most of the equipment, materials, and products are stationary. The mobility and speed of people are low. The likelihood of an unmanned vehicle being hit by a foreign object is low. The system design prevents unmanned vehicles from leaving their lane and colliding with other equipment or materials. With the in-depth development of autonomous driving, image recognition, advanced driver assistance systems, and other technologies, more and more solutions are being proposed for unmanned vehicle systems, such as light detection and ranging (LiDAR), radar, camera, ultrasonic, magnetic stripe, and RFID solutions, and each has its own advantages and disadvantages [8]. Automotive sensors are responsible for the collection and transmission of information. Navigation controllers rely on sensors to recognize changes in the lane environment. Most traditional AGV navigation hardware units comprise on-board processors whose functionality may be limited. In this article, we focus on unmanned vehicle sensing, localization, and navigation control technology under an edge–cloud architecture. We reviewed the related literature and found that:

1. Light detection and ranging (LiDAR) systems collect reflected beams of light to create a 3D image of an object. Although LiDAR is an efficient sensor, it is expensive [9,10].
2. A laser navigation system is suitable for AGV positioning and navigation, but in order to ensure its safe operation, the laser AGV navigation system still needs to use artificial landmarks for positioning and navigation [11].
3. Magnetic stripes and RFID: RFID can store the information of some unmanned vehicles, but metal interferes with RFID, so the technology may fail. In order to improve the positioning accuracy of unmanned vehicles, magnetic nails are used for positioning along lanes [12]. These methods also involve installing hardware (magnets or RFID tags) on the factory floor. In smart manufacturing, installing any type of additional hardware infrastructure in a factory necessitates much follow-up housekeeping, which may impose additional economic costs on the factory.
4. Vision guidance systems [13,14] compare the current image captured by the camera with the stored factory map for navigation, which uses a lot of computing power. This approach is known to cause accuracy issues due to signal reflections in industrial environments. Some methods that use color sensing require other equipment such as RFID tags for positioning, which increase costs [15,16].
5. Ultrasonic signals can be used for unmanned vehicle navigation [17]. However, they are easily disturbed by metal objects in a factory.
6. Wireless networks used for unmanned vehicle localization cause uncertainty in the position of unmanned vehicles due to multipath fading. Another disadvantage of wireless networks is communication delays for autonomous vehicle fleet management [18].
7. When using Wi-Fi, path loss and low available bandwidth (due to the high number of users) cause network latencies. Therefore, it is often necessary to retransmit data packets [19].

8. When using IoT applications, the central computer may not be able to meet the timing requirements, especially in a time-sensitive system. If data are sent to the central computer but a response is not received in time, the consequences are unpredictable [20,21]. In order to distribute the burden of the central computer, edge computing has been developed. Edge computing can simplify hardware requirements at the edge as vehicles can be embedded with a microcomputer to run basic tasks. This strategy helps to reduce the AGV fleet cost [6].

### 1.2. Purpose

Smart factories are the largest market for smart technology, and the combination of precise control, computer vision, and edge computing is the focus of current technological development. This study aims to realize an unmanned vehicle system for factory material transportation, popularize unmanned vehicle systems economically, and solve the problem of manpower shortages in factories. Enterprises pursue not only efficiency but also economic benefits, so the system we develop must meet these needs. Usually, the production line of a factory is reorganized from time to time, and whenever a production line is reorganized, much housekeeping is conducted, e.g., reconfiguring the location of working machines and planning new paths [22,23]. When the work station changes, it is necessary to change the delivery routes and stop points of materials. If the factory uses additional infrastructure for unmanned vehicle navigation, such as magnetic strips, wires, RFID tags, etc., buried in the floor, the construction and maintenance costs of this infrastructure are very high. Thus, factories should move away from these types of installations to accrue economic benefits. As a result, here, we apply innovative methods to quickly adapt to the restructuring needs of factories. An unmanned vehicle cannot stop halfway through the delivery of goods, and it must be ensured that it can complete delivery work independently with fault tolerance. The aisle space in a factory is limited, and two-way traffic is usually present in a single lane. Collisions must be avoided when multiple vehicles are moving at the same time [24–26]. When we integrate the aforementioned technologies for unmanned vehicle systems, there are some main challenges.

1. The real-time image data volume of a car camera is huge (0.4–0.8 Mb/s). Our real-time assumption for this system is that the unmanned vehicle leaves the parking lot and starts to detect images, sends out navigation instructions after image analysis, and then detects images again. The system repeats such work continuously until it returns to the parking lot and waits for the next transportation task. In order for the unmanned vehicle to navigate effectively and safely, the system must complete the navigation operation within 50 ms in most cases. We developed a rapid image recognition algorithm to meet the system requirements.
2. The navigation control of unmanned vehicles requires fast and immediate responses.
3. There are many light disturbances in the factory, such as fluorescent lights, LED lights from instruments, and sunlight, and image recognition must have the ability to eliminate light interference.
4. Image recognition must be able to quickly adapt to factory environments with different lane line colors.
5. Unmanned vehicle systems are designed to maneuver through tight spaces and around obstacles in a factory. They need an avoidance mechanism.
6. We should improve the efficiency of software system maintenance and management.

The system we develop must be able to solve the aforementioned problems and improve the overall operating efficiency.

In this paper, we developed a practical unmanned vehicle system based on edge computing, artificial intelligence, and vision technologies, breaking through the limitation of onboard processors. The novelties of this system are that we have developed an unmanned vehicle system without any additional infrastructure, and we developed a rapid image recognition algorithm for unmanned vehicle systems to improve navigation safety. As a result of our implementation, we found that it is possible to use unmanned vehicles

to transport materials without using expensive sensors and without adding additional infrastructure in the factory, solving the problem of manpower shortage and improving efficiency. The contributions of this work are:

- The unmanned vehicle system we designed is based on the aisle sideline for navigation, without any additional infrastructure, and a low construction cost and easy maintenance. It can support a large number of unmanned vehicle systems in a factory simultaneously.
- We developed a rapid image recognition algorithm that can be used in autonomous vehicles for improving safety.
- We developed an image recognition system with a self-learning mechanism that can analyze parameters with similar colors, improve system analysis decisions, and improve system performance.

## 2. Materials and Methods

There are many navigation solutions for unmanned vehicle systems, and they all face different challenges. For example, light detection and ranging (LiDAR) is expensive [9,10]. Laser AGV navigation systems require the use of landmarks for positioning and navigation [11]. The introduction of such a system necessitates the addition of infrastructure. A visual guidance system compares the current image with a pre-stored map for navigation. In addition to the pre-built image map, these types of systems also require much computing power. This approach can cause accuracy issues due to signal reflections in the factory environment. A method of AGV navigation with color sensors is very useful for AGV navigation due to its low cost, easy installation, and effectiveness during straight-line navigation, but it should be integrated with RFID to assist with positioning [15]. Using vision-based technology for navigation and positioning requires image recognition technology to identify the sidelines.

### 2.1. System Architecture

The main components of our unmanned vehicle include two front wheels. The front wheels are driven by two stepper motors operated independently. A camera and an ultrasonic sensor are installed on the front, and there is also a pocket microcomputer and an Arduino module. The hardware architecture of the unmanned vehicle is shown in Figure 1.

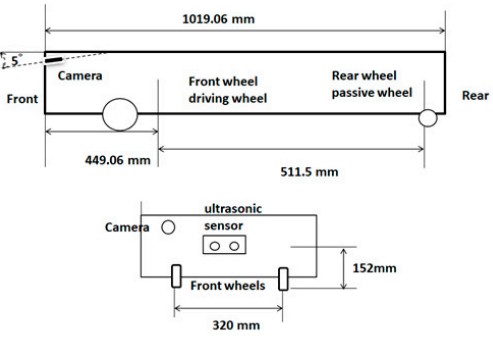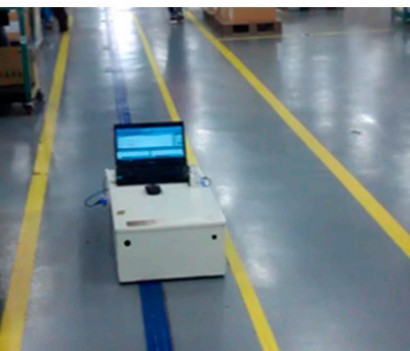

**Figure 1.** Hardware architecture of unmanned vehicle.

We have a camera on the front of the vehicle with the camera lens tilted down 5 degrees to capture images of the floor. An ultrasonic sensor is installed on the front to sense obstacles. The front end of the vehicle has motor-driven wheels. A microcomputer with wireless communication capabilities is placed in the car.

Our system is activated upon receipt of a job order. After a job is assigned, the system generates a routing path, and the navigation system uses this routing path as the skeleton to carry out a transportation task. The vehicle-side microcomputer periodically queries the cloud host through the wireless network whether there is an unfinished task to be executed.

After the microcomputer obtains the list of unfinished jobs, it downloads the routing path. The microcomputer of the vehicle uses the image for analysis and determines the position of the sideline as the basis for navigation control.

### 2.2. AIoT and Edge–Cloud

In this study, we focus on networking, computing power, image recognition, navigation, and security. In Section 1.1, it was pointed out that wireless networks are not suitable for unmanned vehicle navigation due to multipath fading, communication delay, and poor penetration. The indoor network used in this research is mainly used for a small amount of data exchange between the vehicle and the cloud host. With the help of the interconnectivity brought about by the Internet of Things (IoT), as well as its ability to obtain data from equipment, cyber–physical converged systems (CPS) have been introduced. Most of the data from connected devices are collected by sensors equipped with embedded systems, and the data are uploaded over the network to the cloud for storage and processing [27–29]. Intelligent networking involves the introduction of artificial intelligence systems into Internet of Things technologies (AI + IoT = AIoT). AIoT is able to learn from data to generate predictive decisions, which may help to provide higher quality services. Current embedded systems are gradually developing towards miniaturization and intelligence and can integrate sensors for real-time processing. The data received by the sensors do not have to be sent back to the cloud for computation. Instead, real-time processing is performed at edge nodes, which can greatly shorten the time required for data transmission back and forth. Edge computing has the following characteristics:

- It is a distributed computing architecture that transfers the computing of applications and services from central nodes to edge nodes for processing [30].
- It has the ability to support IoT devices on the move.
- It is capable of offloading work from the fog or cloud to edge nodes to improve computational efficiency.

Unmanned vehicles must continue to complete the delivery work as much as possible when the network is disconnected. An edge–cloud collaborative architecture solution is an option that can enhance the system's fault tolerance. The architecture of an edge–cloud system is shown in Figure 2.

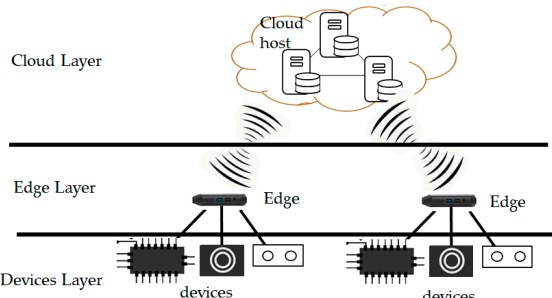

**Figure 2.** Edge–cloud architecture.

In the cloud layer, there are many server hosts connected to support the overall business operation of the enterprise. Important business decisions can be made through big data analysis on cloud servers. The edge host in the cloud layer can connect multiple sensing devices to collect local data, and the sensing data are sent to the edge for analysis and local decision making. According to the system settings, the necessary data are sent back to the cloud for data storage or big data analysis. The device layer contains sensing devices. These sensing devices are responsible for collecting terminal data and sending the data to the edge for data processing. The edge and the cloud can communicate via a wireless network. This architecture can prevent the need for large amounts of data transmission in the network and can also distribute the workload of the cloud or the central host, improve the overall performance of the system, and enhance the scalability of the system.

In order to improve the service efficiency of the unmanned vehicle system, in this paper, we propose and deploy an intelligent unmanned vehicle solution based on AIoT and edge computing technologies. Our unmanned vehicle system uses a pocket microcomputer to improve the execution capability of the vehicles. It has network capabilities, and it can perform complex calculations to make local decisions quickly for navigation. The necessary information is sent to the central system, reducing the need for network transmission. The image data are not transmitted via the network; the network data are only 1/300,000th the size of the images. The novelty of our unmanned vehicle system lies in its improvement of vehicle-side execution capabilities and low network dependence, which make it possible to deploy a large number of unmanned vehicles in factories. The network can be used to remotely maintain and update the software system of the vehicle. We know that the power consumption of network facilities is directly proportional to the amount of data transmitted. The energy consumption per bit of data on the Internet is around 75 microjoules at low access rates and decreases to around 2–4 microjoules at an access rate of 100 Mb/s [31]. For example, a 30 KB image file can save around 0.72 joules of energy consumption. The network transmits less data, the chance of network collision or retransmission is reduced, and the speed and performance of the network are improved. Image processing and navigation are distributed to each vehicle for processing, greatly reducing the burden on the central system.

Unmanned vehicles used in traditional factories are limited by the capabilities of their onboard processors and cannot perform complex computing functions [8]. Our unmanned vehicle system uses a pocket microcomputer that has multiple USB ports and can be used to connect other peripheral devices. Its function is similar to that of a laptop; it uses a Windows system, is small and cheap, and has network communication capabilities. In the future, various software functions can be added at any time to enhance the functions of the unmanned vehicle system. The system can be updated remotely through the network. There is no need to burn the software into the electrically erasable programmable read-only memory (EPROM) separately and then replace the EPROM on the vehicle to update the system, which greatly reduces the system maintenance cost. Such a system architecture can meet the requirements of FOF.

### 2.3. Image Recognition Technology

In order to avoid the extensive housekeeping work caused by the additional installation of hardware infrastructure in a factory, our system uses the factory's existing aisle yellow sidelines as guiding lines for unmanned vehicles without adding any additional infrastructure. A camera in front of the unmanned vehicle captures the vehicle guidance line on the floor and analyzes the position of the guidance line for navigation control. The image recognition strategy of the unmanned vehicle system is based on the color recognition of each pixel. The pixels have red (R), green (G), and blue (B) values of between 0 and 255. The hue (H) value is between −180 and 180, the saturation (S) value is between 0 and 1, and the intensity (I) value is between 0 and 255 [32,33]. There is a lot of light interference in the factory, which causes the color of the pictures taken by the camera to be distorted. The floor color is distorted from gray to blue, as is shown in Figure 3.

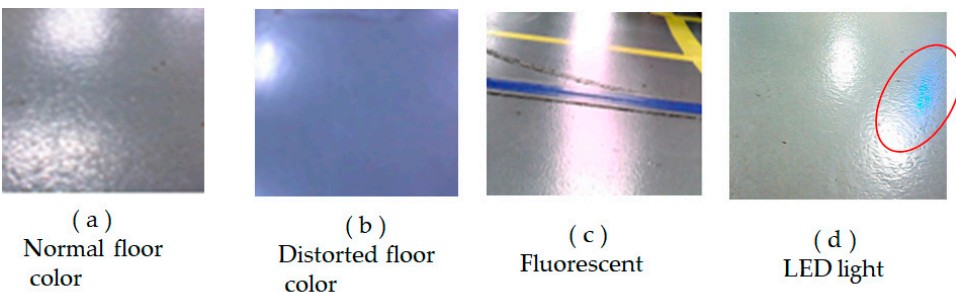

(a) Normal floor color　　　(b) Distorted floor color　　　(c) Fluorescent　　　(d) LED light

**Figure 3.** Distorted floor color.

Figure 3a is a normal image of the floor, and the color of the floor in the picture is the same as the actual color. In Figure 3b, the floor in the image changes to a blue color. If we used a blue line as the vehicle guidance line, this may cause misjudgments. Figure 3c shows light interference caused by fluorescent lights. The yellow line becomes very bright due to the interference, and it cannot be correctly identified as a yellow line. In Figure 3d, the image is disturbed by the LED light of the instrument projected onto the floor. The color distortion causes great problems when it comes to image recognition. Using RGB color model recognition technology, we often cannot identify the floor or the guidance line accurately. Therefore, we used hue–saturation–intensity (HSI) color model technology [34]. HSI color model technology uses three components to represent color. The value of hue is between −180 and 180, the value of saturation is between 0 and 1, and the value of intensity is between 0 and 255, and this provides better distinctions between pixel colors [35–37]. In order to adapt to different factory environments, we designed a self-learning mechanism to analyze the required line color and obtain the parameters required by the system. This eliminates the need for manually testing the photos one by one and recording the hue–saturation–intensity value, reducing the required manpower and shortening the test time. Applying AI's powerful machine learning (ML) [38,39] capabilities to hardware devices enables devices to perform analytics flexibly. ML can usually learn from past data and experiences to find the rules of its operation. In this case, ML involves color pattern training to recognize patterns using examples rather than programming with specific rules. It takes samples from the database to learn and create a hue–saturation–intensity rule for a set of colors and use it to make predictions. We use a full-color template image for the system to compare with the floor image, enabling the system to learn and record the values of the colors. The system automatically distinguishes colors and records related parameters, as shown in Figure 4.

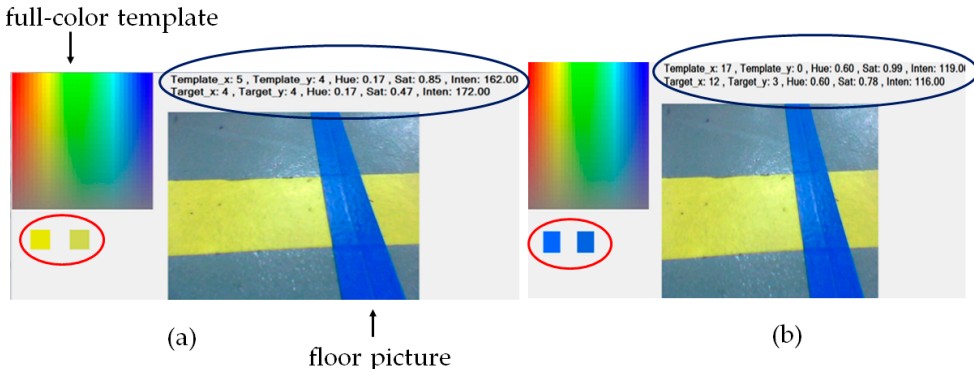

**Figure 4.** Machine learning analysis.

The two yellow blocks in the red circle in Figure 3a are similar in color. The left yellow block in the red circle is from the full-color template, and the right one is from a picture of the yellow line on the floor. The HSI values of the template are: hue (0.17), saturation (0.85), intensity (162), and the values of the floor picture are: hue (0.17), saturation (0.47), intensity (172). This system has found the rules for identifying yellow's hue (0.17), saturation (0.47–0.85), and intensity (162–172) and blue's hue (0.60), saturation (0.78–0.99), and intensity (116–119). The system can also identify various color recognition rules. The system saves the HSI interval values of similar colors as the basis for color judgment, which can eliminate the color distortion caused by external light interference. The recognition system can collect HSI interval values for different colors too, as is shown in Figure 3b for blue. A block diagram of machine learning image recognition analysis process is shown in Figure 5.

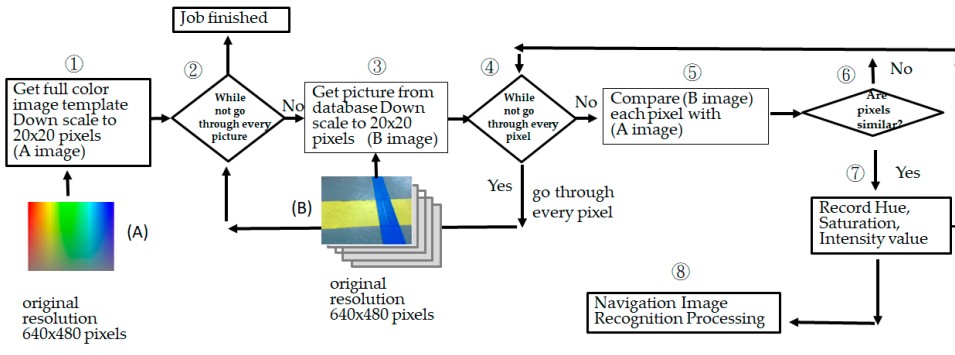

**Figure 5.** Machine learning image recognition analysis process block diagram. Details of the navigation image recognition process is shown in Figure 6.

In the machine learning image recognition analysis process, a full-color template (A) is fed into the image recognition analysis function ①, the resolution is reduced from $640 \times 480$ pixels to $20 \times 20$ pixels, and then the HSI value of each pixel is established and stored in a two-dimensional matrix that has a total of 400 records. The floor image (B) is read from the database using the image recognition analysis function ③, the resolution is reduced from $640 \times 480$ pixels to $20 \times 20$ pixels, and then the HSI value of each pixel is measured and saved in a two-dimensional matrix that has a total of 400 records. Then, the HSI value of every pixel in the two matrices is compared ⑤; if the data are similar, it keeps and records them ⑦. These HSI data are provided to the image recognition module of the vehicle navigation system after passing through each picture in the database ⑧.

*2.4. Sensing and Analysis*

The unmanned vehicle system in this experiment is driven by image recognition. The system uses a web camera as an image capture tool. Each photo captured by the camera has a resolution of $640 \times 480$ pixels and a file size ranging from 20 KB to 40 KB. The amount of image data generated per second is about 400–800 KB. In addition to image recognition, the microcomputer of the unmanned vehicle also performs real-time navigation, and the workload is very heavy. If image recognition and navigation tasks are completed by the cloud host, huge amounts of real-time image data are transmitted through the network. It is not only necessary to keep the network unblocked at all times but also to have sufficient bandwidth, especially when multiple unmanned vehicles are running at the same time. The cloud host may not be able to handle the navigation control of multiple unmanned vehicles at the same time. In this case, the stability of the network becomes critical. The excessive dependence of the system on the network may cause difficulties in the operation of the system, for example, network failure, network congestion, network delay, etc., causing the system to fail. In order to solve the above problems, our system operates under an edge–cloud architecture. After the unmanned vehicle obtains the job instruction and routing path from the cloud host, it can run independently from the cloud host, reducing the system's dependence on the network and reducing the workload of the cloud host. In order to meet the navigation control requirements of unmanned vehicles, we designed a downsizing algorithm to solve this problem. The downsizing and image recognition algorithm pseudo code is as follows:

```
while (y < 480)           // go through y dimension
    while (x < 640)       // go through x dimension
            color = Get_pixel(x,y)
            Hue = Get_Hue(color)
            Saturation = Get_ Saturation (color)
            Intensity = Get_ Intensity (color)
            IF F(Hue, Saturation, Intensity) within margin    // color is matched
                    save line left edge value in 2D array
                    save line top left edge value once
                    save line right edge e value in 2D array
                    save line length
            x = x + 32    // 20 times
    y = y + 24           // 20 times
```

A block diagram of the navigation image recognition process is shown in Figure 6.

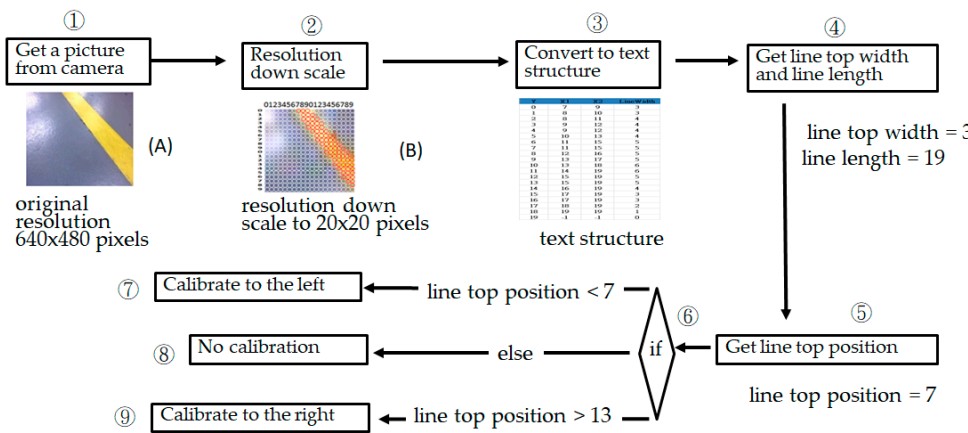

**Figure 6.** Image recognition process block diagram.

After the system obtains the image from the camera ①, it converts the original 640 × 480 resolution image into a 20 × 20 resolution image ②. In ③, the HSI value of each pixel is calculated; if the HSI value of the pixel meets the HSI value of the vehicle guidance line, it records the position of the pixel in the 2D matrix. The guidance line's width and length are obtained from a 2D matrix ④, and the position of the topmost left edge of the guidance line is obtained from the 2D matrix ⑤. If the leftmost edge position of the line is less than 7, the vehicle deviates to the right and must be calibrated to the left ⑦. If the leftmost edge position of the line is greater than 13, the vehicle deviates to the left and must be calibrated to the right ⑨; otherwise, it does nothing.

We compare the color of each pixel, and if the pixel color is similar to the lane edge color, we record the pixel position. We use a two-dimensional matrix to save the position of each pixel of the lane edge, which converts the image into an alphanumeric structure. The converted guideline structure is shown in Table 1.

The y column in Table 1 is the pixel vertical position, the X1 column is the guidance line of the leftmost edge position, and the X2 column is the guidance line of the rightmost edge position. We used the topmost X1 position "7", from the line text structure in Table 1, as the basis for the vehicle's navigation control.

**Table 1.** Line text structure.

| Y | X1 | X2 | Line Width |
|---|----|----|------------|
| 0 | 7 | 9 | 3 |
| 1 | 8 | 10 | 3 |
| 2 | 8 | 11 | 4 |
| 3 | 9 | 12 | 4 |
| 4 | 9 | 12 | 4 |
| 5 | 10 | 13 | 4 |
| 6 | 11 | 15 | 5 |
| 7 | 11 | 15 | 5 |
| 8 | 12 | 16 | 5 |
| 9 | 13 | 17 | 5 |
| 10 | 13 | 18 | 6 |
| 11 | 14 | 19 | 6 |
| 12 | 15 | 19 | 5 |
| 13 | 15 | 19 | 5 |
| 14 | 16 | 19 | 4 |

*2.5. Navigation Control*

The rapid response of the unmanned vehicle system is very important. It should not rely too much on the surrounding facilities, such as the network. In order to improve the fault tolerance of the system, shorten the system response time, and reduce the dependence on the network, the navigation control uses an edge–cloud architecture to reduce the burden on the cloud host. The image data of unmanned vehicles are very large (0.4–0.8 Mb/s), even for a single vehicle. Such data volume relying on wireless network transmission leads to unpredictable results when using multiple vehicles. In order to prevent the system from being unable to operate due to network failure, image recognition and navigation are performed in the unmanned vehicle (edge node). The operation of the system is shown in Figure 7.

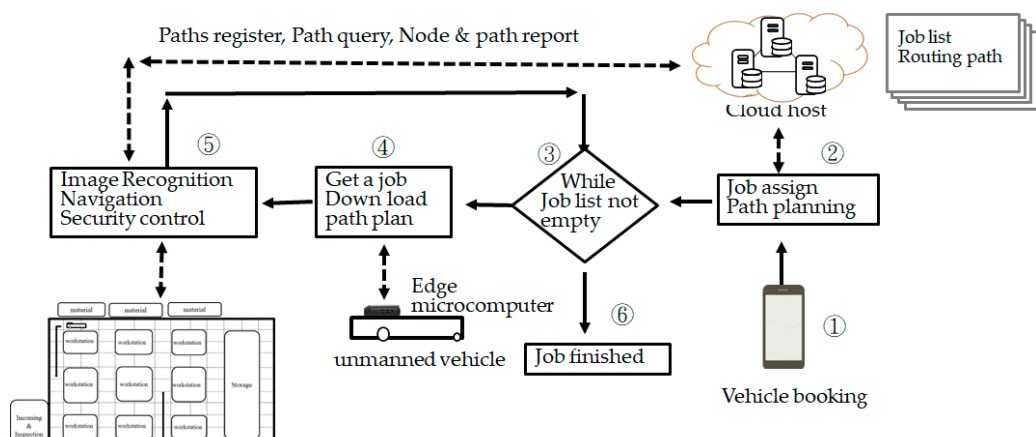

**Figure 7.** System operation.

Vehicle navigation begins with the user booking a car ①. After the user completes the reservation with the cloud system, the cloud system generates a job list and a routing path table ②. The vehicle system periodically checks the cloud system for unfinished tasks ③. If it finds any, it downloads the routing path table ④ and the vehicle system

navigates according to the routing path table. During the navigation process ⑤, the vehicle system registers the path requirements with the cloud system first and checks whether other vehicles are using the same path. If they are, the vehicle system will suspend and wait for five seconds before asking again. If no other vehicles are using the path, the delivery service is started. Navigation continues until the delivery is complete and the AGV is back in the parking lot.

From Figure 6, we can see that there is a loose relationship between our unmanned vehicle and the cloud host, and its time dependence is not strong. The edge microcomputer checks the job list and downloads the route path before the vehicle starts and then registers the required path and reports the upcoming path section or workstation location. Image recognition and navigation control are handled by the edge microcomputer.

The most important aspect of vehicle navigation is controlling the vehicle to move stably in the lane and not to deviate from the lane. Our unmanned vehicle navigates by image recognition technology. A camera is installed at the front of the vehicle to capture images of the lane. The image captured by the camera is located approximately 150 cm in front of the wheel. There is a gap between the wheel and the image location, and the navigation control must include a feedback compensation mechanism. Otherwise, the vehicle is likely to deviate lanes. Let us take a corner turn for illustration purposes. When the vehicle makes a right turn, it turns right without calibration for 3600 milliseconds. During this period, most of the images captured by the camera are messy images at the intersection. We follow the inertial navigation of corner turn and do not perform image recognition during this period. After 3600 milliseconds, the system captures the image again. If the lane image does not exist, it will continue to turn until the lane image is captured. After completing the corner turn, when the camera captures the lane image, the front wheels are actually over-turned, as is shown in Figure 8.

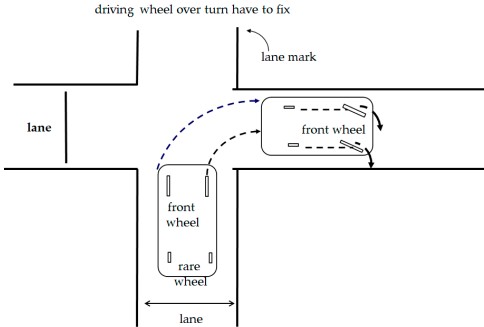

**Figure 8.** Over-turned front wheel.

The direction of the wheel has been over-turned, and the vehicle needs to fix the direction. The compensation formula is as follows:

$$\text{Fix\_Time} = N * (\Sigma \text{ Turning\_Time})$$

where N = 0.25 − 0.8 and Fix_Tim is the compensation time for the wheel to be brought back parallel to the body.

Unmanned vehicles undergo direction calibration during driving. If the direction calibration turns too far, it will cause the vehicle to travel in an S-shape. This situation may cause the rack to overturn, which is dangerous. We used a fuzzy control method [40] to prevent this situation during navigation. We expanded the judgment range of straight-line driving, narrowed the calibration range of each calibration, and performed half the calibration operations each time. This calibration method can help avoid S-type driving.

Most of the objects in the factory are stationary, and some moving objects do not move fast, so the chance of a vehicle colliding with a foreign object is small. The anti-collision mechanism is mainly meant to avoid vehicle head-on collisions or deadlock. As is shown in Figure 7, during navigation, the unmanned vehicle registers the path with the cloud

system and reports the next path to be used. If there is a conflict, it stops and avoids it using the conflicting path. We utilize such a mechanism to prevent two vehicles from colliding or deadlocking. In order to prevent the unmanned vehicle from bumping into people or objects, we use ultrasonic sensing at the front of the vehicle. When the vehicle senses that there is an object within 60 cm in front, it will stop and emit a warning sound.

## 3. Results

In this section, we present three categories of results to evaluate this system: (1) evaluate the performance of machine learning image recognition analysis; (2) evaluate the performance of image recognition; (3) navigation control test results.

### 3.1. Machine Learning Image Analysis Results

The system completes the image analysis and records relevant parameters in a few seconds. The results are ready to use and fast. The machine learning time consumption for each picture is shown in Table 2.

**Table 2.** Machine learning time.

| Platform | Image Size | Analysis Time (Milliseconds) |
|---|---|---|
| Intel Core i7-3540M CPU: 3 GHz RAM: 8 GB OS: Windows 7 | Picture 1 20 × 20 | 3411.92 |
| | Picture 2 20 × 20 | 3565.50 |
| | Picture 3 20 × 20 | 3166.70 |

As seen in Table 2, we tested the self-learning system with three pictures, and the time required for each image is about four seconds. Our design can improve the execution speed by setting the learning region externally.

### 3.2. Image Recognition Test Results

The images are taken with a camera that captures 640 × 480 pixels. We use raw images for image analysis processing, which takes about 195.07 milliseconds per image. Thus, image analysis and processing take too long. For navigational safety requirements, the analysis time per picture must be less than 50 milliseconds according to the speed of our unmanned vehicle. We downscaled the images to 20 × 20 pixels, which takes about 0.51 milliseconds per image. The image recognition process is shown in Figure 6; we compared images with different resolutions, and the results are shown in Table 3.

**Table 3.** Image analysis time.

| Platform | Image Resolution | Analysis Time (Milliseconds) | Meet Requirement (50 Milliseconds<) |
|---|---|---|---|
| Intel Core i7-3540M CPU: 3 GHz RAM: 8 GB OS: Windows 7 | 640 × 480 | ≈195.07 | ✘ |
| | 100 × 100 | ≈6.35 | ✔ |
| | 20 × 20 | ≈0.51 | ✔ |

Our system moves about 70 centimeters per second at normal speed. In order to meet driving safety requirements, the recognition time of each image cannot exceed 50 milliseconds. Table 3 shows that the image recognition time using the original image resolution of 640 × 480 pixels is 195+ milliseconds, which is not acceptable. When resizing the resolution to 20 × 20 pixels, the image recognition time is 0.51 milliseconds, and the image recognition efficiency was improved by 282 times. This system uses color as the basis for

target recognition, and reducing the image size from $640 \times 480$ to $20 \times 20$ does not change the pixel color. Our algorithm can clearly express the outline position of the image with structured numbers.

### 3.3. Navigation Control Test Results

We tested the unmanned vehicle system in a factory smoothly. Each production line had 15 stations within a circular area of about 180 m. It takes about 6 min for the unmanned vehicle to circle around. The system test results can successfully prevent multi-vehicle collision or deadlock during navigation. If there is a conflict at crossroads, the vehicle stops and waits, just like traffic lights at crossroads.

## 4. Discussion

From the actual system test results, we found that our technology can be applied to unmanned vehicles in factories to assist in the delivery of materials. Therefore, this method is worth discussing.

1. System scalability

The system is implemented based on an edge–cloud architecture. The vehicle-side microcomputer is responsible for image recognition and navigation control. A small amount of data are sent to the cloud host during driving. Under this mechanism, the system can support a large number of unmanned vehicles driving in a factory at the same time. When an unmanned vehicle is moving, it only uploads a few bytes of data to the cloud host, which is about 1/300,000th the size of the image data ($640 \times 480$ image data). The demand for network bandwidth is greatly reduced, and thus the network is able to support multiple vehicles at the same time. Assuming that the network can only provide one vehicle to transmit image data, and our system only transmits 1/300,000th of the data volume, it can theoretically support 300,000 vehicles, but we know that when the network transmission data increase, the performance decreases. Regardless, the system has scalability. This system is suitable for the large-scale deployment of unmanned vehicles in factories.

2. Cost effectiveness

Our system uses the factory's existing colored aisle sidelines for navigation and does not require any new infrastructure. Other types of navigation methods, such as those mentioned in Section 1.1, require additional positioning facilities, increasing the construction and maintenance costs. The installation and maintenance costs of this system are low, and its maintenance is convenient. A comparison of our system with other systems is shown in Table 4.

**Table 4.** Comparison with other systems.

| Type | Additional Facilities | Construction Costs | Maintenance Costs |
|:---:|:---:|:---:|:---:|
| System with LiDAR sensor | No | High | Low |
| System with Laser sensor | Yes | High | High |
| System with magnetic stripes and RFID | Yes | High | High |
| System with vision guidance sensor | No | High | High |
| Our system with camera | No | Low | Low |

LiDAR is expensive, and the construction cost of the system using LiDAR as a navigation sensor is high [9,10]. Laser navigation systems use artificial landmarks for localization and navigation, and magnetic pins for lane positioning [11,12]. Magnetic stripe and RFID

methods involve mounting hardware (magnets or RFID tags) on the factory floor [13]. The construction cost and maintenance cost of these navigation systems requiring the use of additional positioning facilities are high. Vision guidance compares the current image captured by the camera with the stored factory map for navigation, requiring pre-built image maps [14,15], and most systems are costly to build and maintain. In contrast, our system uses a standard camera, uses the factory's existing aisle edges for navigation, and can navigate smoothly without the need to add any infrastructure. The construction cost and maintenance of the unmanned vehicle system are relatively low, making it a good choice for industrial introduction.

3. IoT short message service (SMS) centralized collision avoidance

Most of the objects in the factory are stationary, and some moving objects do not move quickly. Therefore, it is not necessary to install roadside sensing facilities. Furthermore, installing any kind of additional hardware infrastructure in a factory leads to many subsequent management issues, which may impose additional economic costs on the factory. We propose an IoT SMS centralized collision avoidance system, as is shown in Figure 7. During navigation, the unmanned vehicle registers the path with the cloud system and reports the next path to be used. If there is a conflict, it stops and avoids it using the conflicting path.

4. Large image to small structured text conversion

A 20–40 KB (640 × 480) image file is quickly (about 2 ms) converted into 120-byte structured text by our algorithm to achieve navigation control.

5. System maintenance service without tears

Traditional AGV navigation hardware units have on-board processors, and their software systems are stored on the electrically erasable programmable read-only memory (EPROM). If the system needs to be updated, the engineers must replace the hardware on-site, which is very time-consuming and costly. Our vehicle microcomputer connects to the network. When the system needs to be updated, this can be achieved remotely through the network, which is very convenient and can maintain the system in a time-efficient manner, saving manpower, time, and money. Across regions or countries, this is important, as it greatly reduces the system maintenance costs and helps to meet the requirements of FOF.

Autonomous vehicles are popular products at present. Many technologies related to autonomous vehicles are constantly being researched. The technology proposed here is a form of underlying technology. These research results and methods may provide reference values or can be integrated with other technologies.

## 5. Conclusions

In this experiment, we integrated image recognition and artificial intelligence IoT technology under an edge–cloud architecture to realize an unmanned vehicle for use in a factory. In addition to those discussed in Section 4, the system has the following benefits:

- The image recognition system has a self-learning mechanism that can analyze the similar color parameters of lane lines in order to improve the system analysis and decision-making parameters and the system performance.
- The image recognition algorithm can remove light interference.
- Our system takes about 50 milliseconds per calibration operation, and the vehicle moves about 3.5 cm per calibration. The images are taken about 100 cm in front of the car, and there are about 28 chances for the car to calibrate its direction before it exits its lane.

Today, with the vigorous development of unmanned vehicles, the safety and stability of autonomous driving are very important. The integration of various advanced sensing devices to improve system performance is inevitable in the future.

## 6. Patents

This research obtained the following Taiwan invention patent:
Certificate number: 1671609
Patent name: Automatic Guided Vehicle and Control Method Thereof.

**Author Contributions:** Conceptualization, Y.-H.K. and E.H.-K.W.; methodology, Y.-H.K. and E.H.-K.W.; software, Y.-H.K.; validation, E.H.-K.W.; formal analysis, E.H.-K.W.; investigation, Y.-H.K.; resources, Y.-H.K.; data curation, Y.-H.K.; writing—original draft preparation, Y.-H.K.; writing—review and editing, Y.-H.K. and E.H.-K.W.; visualization, Y.-H.K.; supervision, E.H.-K.W.; project administration, E.H.-K.W. All authors have read and agreed to the published version of the manuscript.

**Funding:** This study received no external funding.

**Data Availability Statement:** The datasets generated and/or analyzed during the current study are available at the following drive.google.com repository: https://drive.google.com/drive/folders/1SdvlYTA5oHnHLO60ahJaUG2fa9gyppzx (accessed on 1 January 2023).

**Acknowledgments:** This work was supported by the Institute for Information Industry and SHA YANG YE, Inc.

**Conflicts of Interest:** The authors declare no conflict of interest.

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
