# Peer review of "Advanced, Innovative AIoT and Edge Computing for Unmanned Vehicle Systems in Factories"

_electronics, doi:10.3390/electronics12081843_

Round 1
Reviewer 1 Report
The paper needs to present it's novelty at the beginning. The abstract says nothing about the aims/contribution of the paper.
The paper has AIoT in the title, but nothing in the papers actually presents anything novel about the intelligent device - characteristics e.g., scalability, power consumption, speed under various conditions.
Overall the presented work is sound and analytically correct, but it seems to be an elaborate version of a simple line follower robot. I may be missing the novelty because the main challenge in what the robot is doing is not mentioned in details.
Author Response
Dear reviewer
Thank you, my response is attached.Point 1: The paper needs to present it's novelty at the beginning. The abstract says nothing about the aims/contribution of the paper.
Response 1: We added novelty, aims and contribution in Abstract
Point 2: The paper has AIoT in the title, but nothing in the papers actually presents anything novel about the intelligent device - characteristics e.g., scalability, power consumption, speed under various conditions.
Response 2: We added novelty, scalability, power consumption, speed in 2.1 AIoT and Edge–Cloud Technology section.
Point 3: Overall the presented work is sound and analytically correct, but it seems to be an elaborate version of a simple line follower robot. I may be missing the novelty because the main challenge in what the robot is doing is not mentioned in details.
Response 3: We supplement the main challenges in Section 1.2
Best regards
Yen-Hui Kuo
Reviewer 2 Report
The manuscript “Advanced AIoT and Edge Computing Innovative Unmanned Vehicle Systems for Factories” proposes an advanced unmanned vehicle system based on edge computing, artificial intelligence, and vision technologies, breaking through the limitation of onboard processors. The manuscript has some major issues listed as follows:
1. The topic is exciting and trendy. Therefore, the procedure presentation is quite good.
2. The contribution(s) of this work should be listed as bullets list at the end of the introduction section.
3. What do the authors mean by applying AI's powerful machine learning? This must be explained in the manuscript clearly.
4. There is no information about the applied machine learning algorithm.
5. In Table 2, why are there 3 analysis times?
6. Reducing the image size from 640x480 to 20x20 significantly reduces the analysis time, but what about the quality? Is this considered an issue, and how do the authors deal with it?
7. Is this system can be considered a real-time system? For being a real-time system, what assumptions are needed?
8. The author did not compare the proposed work with any related work to showcase the performance of the proposed work among the literature.
9. There are many spelling mistakes inside the paper. A complete revision must occur for the paper.
Author Response
Dear reviewer
Thank you, my response is attached.Point 1: The topic is exciting and trendy. Therefore, the procedure presentation is quite good.
Response 1: Thanks a lot.
Point 2: The contribution(s) of this work should be listed as bullets list at the end of the introduction section.
Response 2: We listed contribution as bullets list at (1.2 section) the end of the introduction section
Point 3: What do the authors mean by applying AI's powerful machine learning? This must be explained in the manuscript clearly.
Response 3: We supplement machine learning in Section 2.2. Image Recognition Technology
Point 4: There is no information about the applied machine learning algorithm.
Response 4: We explain how to apply the data of machine learning algorithm in 2.2. Image Recognition Technology Figure 3.
Point 5: In Table 2, why are there 3 analysis times?
Response 5: we test the three pictures separately and we modified the table to clearly express.
Point 6: Reducing the image size from 640x480 to 20x20 significantly reduces the analysis time, but what about the quality? Is this considered an issue, and how do the authors deal with it?
Response 6: In 3.3. Image Recognition Test Results sectio Table 3, we explain how to deal with the issue of resolution reduction without affecting image judgment.
Point 7: Is this system can be considered a real-time system? For being a real-time system, what assumptions are needed?
Response 7: We supplemented the real-time operation requirements of this system in section 1.2. Purpose
Point 8: The author did not compare the proposed work with any related work to showcase the performance of the proposed work among the literature.
Response 8: In Section 4. Discussion, we use a table to supplement the differences with other types of sensors
Point 9: There are many spelling mistakes inside the paper. A complete revision must occur for the paper.
Response 9: We adopt Electronics Editorial Office Suggestions, and request editing services listed at https://www.mdpi.com/authors/english assisted in the modification, this version is a modified version.
Best regards
Yen-Hui Kuo
Round 2
Reviewer 1 Report
The paper still has serious faults.
It is extremely unorganised.
What is the relevance of Figure 9 in context of the proposed indoors method?
Abstracts cannot have dot points and the text is simply repeated in line 170 onwards.
Where is the image recognition or downsizing algorithm?
From the general descriptions, I can guess on what the robot is doing. And I think form the shown images, the presented results are achievable.
But the claimed main contributions are simply not in the paper and there is no way to even imagine what the solutions could be.
Author Response
Point 1: The paper still has serious faults.
Response 1: We supplement the content according to various Comments and Suggestions
Point 2: It is extremely unorganized.
Response 2: We reorganized section 2.1. System Architecture, section 3.1. Machine learning image analysis results, section 3.2. Image Recognition Test Results, section 3.3. Navigation Control Test Results
Point 3: What is the relevance of Figure 9 in context of the proposed indoors method?
Response 3: We removed Figure 9 as suggested to avoid readers mistaking us for use of other roadside facilities.
Point 4: Abstracts cannot have dot points and the text is simply repeated in line 170 onwards.
Response 4: We adjusted the Abstract as suggested, rather than simply repeating it.
Point 5: Where is the image recognition or downsizing algorithm?
Response 5: Added downsizing and image recognition algorithm pseudo code in Section 2.4.
Point 6: From the general descriptions, I can guess on what the robot is doing. And I think form the shown images, the presented results are achievable.
Response 6: We tested the system in a factory smoothly.
Point7: But the claimed main contributions are simply not in the paper and there is no way to even imagine what the solutions could be.
Response 7: The core contribution of this system is that, the system can navigate smoothly without expensive sensors and without any additional infrastructure and it can simultaneously support a large number of unmanned vehicle systems in a factory. In Section 2.3. it is stated that we use the existing aisle edge of the factory as the basis for navigation without any additional infrastructure. In Section 2.4. the image recognition method is described. We supplement the scalability statement of the system in item 1 of Section 4.

Reviewer 2 Report
Thanks to the authors for addressing my comments. The manuscript has some major issues listed as follows:
1. The contribution(s) of this work should be listed as bullets list at the end of the introduction section, NOT at the end of the abstract.
2. Is this system can be considered a real-time system? For being a real-time system, what assumptions are needed? This question has not been answered yet. The author only addressed the problem without explaining how to achieve this in their model.
3. Adjust the place of Figure 3 so that it does not intersect with the paragraph.
Author Response
Point 1: The contribution(s) of this work should be listed as bullets list at the end of the introduction section, NOT at the end of the abstract.
Response 1: We have adjusted the abstract as suggested
Point 2: Is this system can be considered a real-time system? For being a real-time system, what assumptions are needed? This question has not been answered yet. The author only addressed the problem without explaining how to achieve this in their model.
Response 2: we add “Our real-time assumption for this system is that the unmanned vehicle leaves the parking lot and starts to detect images, sends out navigation instructions after image analysis, and then detects images again. The system repeats such work continuously until it returns to the parking lot and waits for the next transportation task. In order for the unmanned vehicle to navigate effectively and safely, the system must complete the navigation operation within 50ms in most cases. We developed a rapid image recognition algorithm to meet the system requirements. “ in item 1 of Section 1.2 Purpose.
Point 3: Adjust the place of Figure 3 so that it does not intersect with the paragraph.
Response 3: We have adjusted as suggested and go through every Figure.

Round 3
Reviewer 1 Report
Now the paper has all the parts needed, although I still do not see the novelty in the paper's work over line follower robots as it is not clearly written or the work is not compared to similar papers.
Line 569 is unreadable, and the data link in https://drive.google.com/drive/folders/1SdvlYTA5oHnHLO60ahJaUG2fa9gyppzx requires permissions which I cannot ask for without revealing my identity.
This kind of lazy casual approach to writing this paper lowers the reader's confidence.
Nevertheless, I think if the paper is edited thoroughly, it can be published.
Author Response
Point 1: Now the paper has all the parts needed, although I still do not see the novelty in the paper's work over line follower robots as it is not clearly written or the work is not compared to similar papers.
Response 1: We supplement the novelty in the Abstract. “The novelties of this system are that we have developed an unmanned vehicle system without any additional infrastructure, and we developed a rapid image recognition algorithm for unmanned vehicle systems to improve navigation safety.” We supplement a comparison of our system with other systems is shown in Table 4, and explain in Section 4, item 2.
Point 2: Line 569 is unreadable, and the data link in https://drive.google.com/drive/folders/1SdvlYTA5oHnHLO60ahJaUG2fa9gyppzx requires permissions which I cannot ask for without revealing my identity.
Response 2: The link has been changed so that any internet user with the link can view without asking for permission.
Point 3: This kind of lazy casual approach to writing this paper lowers the reader's confidence.
Response 3: Thanks for your suggestions on this paper
Point 4: Nevertheless, I think if the paper is edited thoroughly, it can be published.
Response 4: Thanks for your suggestions on this paper

Reviewer 2 Report
Thanks to the authors for addressing the comments.
There are no more comments.
Author Response
Point 1: Thanks to the authors for addressing the comments. There are no more comments.
Response 1: Thank you.